# Simultaneous Measurement and Distribution Analysis of Urinary Nicotine, Cotinine, Trans-3′-Hydroxycotinine, Nornicotine, Anabasine, and Total Nicotine Equivalents in a Large Korean Population

**DOI:** 10.3390/molecules28237685

**Published:** 2023-11-21

**Authors:** Hyun-Seung Lee, Mi-Ryung Chun, Soo-Youn Lee

**Affiliations:** 1Department of Laboratory Medicine, School of Medicine, Wonkwang University, 895 Muwang-ro, Iksan-si 54538, Jeollabuk-do, Republic of Korea; ctmasaru@gmail.com; 2Department of Laboratory Medicine and Genetics, Samsung Medical Center, School of Medicine, Sungkyunkwan University, 81 Irwon-ro, Gangnam-gu, Seoul 06351, Republic of Korea; miryung.chun@samsung.com; 3Department of Clinical Pharmacology and Therapeutics, Samsung Medical Center, School of Medicine, Sungkyunkwan University, 81 Irwon-ro, Gangnam-gu, Seoul 06351, Republic of Korea

**Keywords:** cotinine, nicotine, toxicology, surveillance and monitoring

## Abstract

Measurement of multiple nicotine metabolites and total nicotine equivalents (TNE) might be a more reliable strategy for tobacco exposure verification than measuring single urinary cotinine alone. We simultaneously measured nicotine, cotinine, 3-OH cotinine, nornicotine, and anabasine using 19,874 urine samples collected from the Korean National Health and Nutrition Examination Survey. Of all samples, 18.6% were positive for cotinine, 17.4% for nicotine, 17.3% for nornicotine, 17.6% for 3-OH cotinine, and 13.2% for anabasine. Of the cotinine negative samples, less than 0.3% were positive for all nicotine metabolites, but not for anabasine (5.7%). The agreement of the classification of smoking status by cotinine combined with nicotine metabolites was 0.982–0.994 (Cohen’s kappa). TNE3 (the molar sum of urinary nicotine, cotinine, and 3-OH cotinine) was most strongly correlated with cotinine compared to the other nicotine metabolites; however, anabasine was less strongly correlated with other biomarkers. Among anabasine-positive samples, 30% were negative for nicotine or its metabolites, and 25% were undetectable. Our study shows that the single measurement of urinary cotinine is simple and has a comparable classification of smoking status to differentiate between current smokers and non-smokers relative to the measurement of multiple nicotine metabolites. However, measurement of multiple nicotine metabolites and TNE3 could be useful for monitoring exposure to low-level or secondhand smoke exposure and for determining individual differences in nicotine metabolism. Geometric or cultural factors should be considered for the differentiation of tobacco use from patients with nicotine replacement therapy by anabasine.

## 1. Introduction

The measurement of biomarkers related to tobacco exposure is the most widely used objective method for smoking status verification [1]. Cotinine, which is the major metabolite of nicotine [2,3], is the most widely used biomarker for biochemical verification of smoking status [4]. Urinary cotinine is a non-invasive biomarker with diagnostic performance similar to that of serum cotinine [2,5]. Therefore, the Korean National Health and Nutrition Examination Survey (KNHANES) has used urinary cotinine as a biomarker to verify smoking status and to monitor population exposure to tobacco over time since 2008 [6,7].

Cotinine concentration can be affected by ethnicity, genetics, medical conditions, and environmental factors [3,8,9]. The half-life of cotinine varies from 8 h to over 30 h due to these factors [4]. CYP2A6 is the major enzyme responsible for the metabolism of nicotine to cotinine, which is further metabolized to trans-3′-hydroxycotinine (3-OH cotinine) as well as 5′-diphospho-glucuronyltransferase (UGT) [9,10]. Considering the inter-individual variabilities in cotinine metabolism due to many genetic and non-genetic factors, the measurement of multiple nicotine metabolites is more reliable for the assessment of tobacco exposure compared to a single urinary cotinine measurement [11,12]. In addition, total nicotine equivalents (TNE), the sum of most or all nicotine metabolites in urine [4], is suggested to be the most accurate biomarker of nicotine intake. TNE has been demonstrated to be highly correlated with nicotine intake at known doses, while single cotinine alone showed a weaker correlation in people with reduced CYP2A6 metabolic activity [13]. TNE has been reported to be effective in assessing changes in nicotine intake over time within subjects, while cotinine is less accurate for this due to differences in metabolism [4,13]. Therefore, the measurement of multiple nicotine metabolites and TNE in urine might be a more reliable strategy for evaluating tobacco exposure [13].

Lower urinary biomarker concentrations due to decreased metabolic activity and low-level or secondhand smoke exposure cannot be detected with a single biomarker. Several studies have reported the usefulness of measuring multiple nicotine metabolites and TNE for evaluation of secondhand exposure or smoking cessation compared to single cotinine alone [14,15,16]. However, research on the East Asian population is still lacking [14], although these populations have a higher prevalence of reduced CYP2A6 metabolic activity compared to other ethnicities [13,17].

Urinary biomarkers can be analyzed by a variety of analytical methods, including immunoassay, gas chromatography (GC), GC-mass spectrometry (GC-MS), liquid chromatography (LC), and liquid chromatography with tandem mass spectrometry (LC-MS/MS) [4]. The measurement of single urinary cotinine is widely available in laboratories and has been used for population screening [4]. However, measurement of multiple nicotine metabolites and TNE is only available at a limited number of laboratories, as it is more expensive and the procedure is more complex [4]. Therefore, a comparison of the usefulness of single urinary cotinine versus multiple nicotine metabolites in a large population for evaluating tobacco exposure is important. Large-scale data on multiple marker measurements in comparison to a single cotinine assay would be helpful in developing an effective strategy for population monitoring for tobacco exposure.

Recently, our group reported a novel LC–MS/MS method for the detection of anabasine, nicotine, and three nicotine metabolites consisting of cotinine, 3-OH cotinine, and nornicotine (the molar sum of nicotine, cotinine, and 3-OH cotinine known as TNE3) in urine, with a simple sample preparation technique [18]. TNE3 is known to be highly correlated with the sum of all nicotine metabolites [4,13], and the simple and economical sample preparation procedure and short run times might be useful for analyzing a large number of samples, such as a national survey. In the current study, we evaluated six biomarkers related to smoking exposure: nicotine, cotinine, 3-OH cotinine, nornicotine, anabasine, and urinary TNE3, using approximately 20,000 urine samples obtained from KNHANES. The aim of our study was to assess the agreement between single urinary cotinine and multiple nicotine metabolites in a large number of samples from the national survey. For this, we compared biomarker-based strategies between single cotinine and multiple nicotine metabolites for monitoring population exposure to tobacco.

## 2. Results

### 2.1. Results of the LC–MS/MS Analysis

Results of the LC–MS/MS analysis are shown in Table 1. Samples categorized as “not detected” for each analyte comprised 76.4–81.7% of the 19,894 samples. The proportions of positive samples with a concentration above the cut-off [19] were 18.6% (3682/19,894), 17.4% (3460/19,894), 17.3% (3434/19,894), 16.6 (3308/19,894), and 13.2% (2624/19,894) for cotinine, nicotine, nornicotine, 3-OH cotinine, and anabasine, respectively, with concentrations ranging up to 4907.4 ng/mL, 13,806.4 ng/mL, 766.5 ng/mL, 39,604.5 ng/mL, and 161.1 ng/mL, respectively. TNE3 concentrations varied from <0.1 µmol/L to 268.5 µmol/L.

### 2.2. The Distribution of Six Urinary Biomarkers

The distribution of six urinary biomarkers from all samples is illustrated in Figure 1. On visual inspection of the histogram, urinary cotinine, nicotine, nornicotine, 3-OH cotinine, and TNE3 had a bimodal distribution (Figure 1A–D,F), whereas anabasine did not (Figure 1E). The cut-off values of urinary cotinine, nicotine, nornicotine, and 3-OH cotinine were located between two peaks of distribution, whereas that of anabasine was not (Figure 1E).

### 2.3. A Comparison of the Measurement Status between the Five Urinary Biomarkers 

A comparison of the detection status of five urinary biomarkers related to smoking exposure is presented in Table 2. Of the 4679 samples with detectable urinary cotinine, other biomarkers were detected in 65.8–88.0%. Of the 15,215 samples without detectable urinary cotinine, anabasine was detected in 7.9%, whereas nicotine and its metabolites were detected in 0.1–2.4%.

A comparison of the classification of smoking status based on five biomarkers is presented in Table 3. Of the 3682 positive samples for urinary cotinine, 71.7% were positive for anabasine, whereas 93.3–96.4% were positive for nicotine and its metabolites. Of the 16,212 negative samples for urinary cotinine, less than 0.3% of the samples were negative for all biomarkers except anabasine (5.7%, 925/16,212).

### 2.4. Agreement of the Measurement Status between the Five Urinary Biomarkers 

Nicotine and its metabolites showed a very high agreement for detection status (Cohen’s kappa; 0.838 to 0.973) (Table 2) and showed higher agreement for classification of smoking status (0.948 to 0.978) (Table 3). However, anabasine showed lower agreement for detection status (0.597 to 0.683) (Table 2) or classification of smoking status (0.657 to 0.693) (Table 3) compared to the other nicotine metabolites.

### 2.5. Correlations between the Six Urinary Biomarkers

Correlations between the six urinary biomarkers are presented in Appendix A. Positive correlations were observed between the biomarkers. Urinary cotinine showed a very high correlation with other nicotine metabolites (range: 0.8994 to 0.9630; Appendix A). Among them, TNE3 was most strongly correlated with cotinine compared to other nicotine metabolites (Appendix A). However, urinary anabasine was less strongly correlated with other biomarkers. (anabasine with cotinine, r = 0.6674; anabasine with nicotine, r = 0.6835; anabasine with nornicotine, r = 0.7319; anabasine with 3-OH cotinine, r = 0.6798; anabasine with TNE3, r = 0.6514; and *p* < 0.0001 for all).

## 3. Discussion

The study by Benowitz et al. [13] showed that TNE3, the molar sum of urinary nicotine, cotinine, and 3-OH cotinine, is highly correlated with the sum of nicotine and all of its metabolites, as well as serum cotinine concentration. Therefore, TNE3 is one of the most useful biomarkers for assessing the daily intake of nicotine. However, the diagnostic performance of TNE3 to differentiate between current smokers and non-smokers is still unclear [4].

In the current study, there does not appear to be a remarkable difference in the classification of smoking status based on urinary biomarkers between urinary cotinine and TNE3, although the criterion for TNE3 has not yet been established. When positive samples of nicotine metabolites were measured, differences of 0.1–0.3% were found compared to urinary cotinine alone (Table 3). The degree of concordance in the classification of smoking status by urinary cotinine combined with nicotine metabolites compared with urinary cotinine alone was almost perfect (Cohen’s kappa 0.982–0.994; Appendix A). Therefore, our study shows that a single measurement of urinary cotinine, compared to multiple measurements of nicotine metabolites, is more cost-effective to differentiate between current smokers and non-current smokers.

On the other hand, urinary nicotine and 3-OH cotinine were detected at 2.4% and 0.7%, respectively, in samples with undetectable urinary cotinine, and the degree of concordance for the detection status among nicotine and its metabolites was lower compared to the classification of smoking status based on urinary biomarkers (Table 2 and Table 3). A single biomarker was detectable at 9.0%, and partial biomarkers (two or three) were detectable at 4.1%, whereas 5.1% was positive for a single biomarker and 1.5% was positive for two or three biomarkers for the classification of smoking status based on urinary biomarkers. (Table 4 and Appendix A). When excluding anabasine, one to three biomarkers were detectable in 6.7% (1345 cases). One hundred cases (0.6%) were positive for a single biomarker, and 271 cases (1.4%) were positive for two or three markers for the classification of smoking status. Discrepancies in the detection status between nicotine metabolites were mostly observed around the lower limit of measuring interval (LLMI) below the cut-off values.

TNE3 showed a different value of 3.0% in those samples compared to the value for the sum of the LLMI of urinary nicotine, cotinine, and 3-OH cotinine. These results could be explained by individual differences in nicotine metabolism [4], as well as differences in the half-lives of each biomarker [20], and sampling time [21,22]. Among the positive samples for urinary cotinine, the ratio of urinary 3-OH cotinine to cotinine (NMR) [23] varied from 0.00 to 34.16 in our study (Appendix A). A person who had reduced CYP2A6 or uridine UGT activity would slowly metabolize nicotine, resulting in more nicotine excreted as the nicotine form [13,24]. If a person had increased enzyme activity, more nicotine and cotinine would be excreted in the 3-OH cotinine form. According to a previous study for secondhand smoke exposure in the KNHANES survey [25], single urinary cotinine was a less sensitive biomarker for distinguishing secondhand smoke exposure from non-current smokers because urinary cotinine was unmeasurable in 10.2% of participants who self-reported a history of secondhand smoke exposure. Collectively, our study suggests that measurement of multiple nicotine metabolites and TNE3 might be helpful for monitoring exposure to secondhand or thirdhand smoke, compared to measurement of single urinary cotinine.

Anabasine is one of the minor tobacco alkaloids [26]. Anabasine is a useful biomarker for tobacco use in people using nicotine patches or gum because it is present in very small amounts in pharmaceutical-grade nicotine products [4,27]. In 2002, the cut-off of 2 ng/mL for urinary anabasine was defined as the criterion for distinguishing between current smokers and non-smokers [27]. However, several studies [28,29,30] have reported lower sensitivity of urinary anabasine compared to serum cotinine when using the current cut-off value. Except for samples with values below the LLMI, urinary anabasine was log-normally distributed, whereas urinary cotinine, nicotine, nornicotine, 3-OH cotinine, and TNE3 were bimodally distributed in the current study (Figure 1). This finding for the distribution of urinary anabasine is similar to those of a previous study using KNHANES study data between 2013 and 2014 [31].

Recently, Colsoul et al. suggested that 0.236 ng/mL of anabasine appears to better distinguish between current smokers and non-smokers than the cut-off value of 2 ng/mL [31]. In the current study, 19.5% of samples showed values below the LLMI for urinary anabasine, among the positive samples for urinary cotinine. This finding is in line with a recent study [32], which showed improved sensitivity for distinguishing between current smokers and non-smokers when using 0.236 ng/mL as the novel cut-off value for urinary anabasine. Although the LLMI of anabasine in the current study was higher than 0.236 ng/mL, our data support the adequacy of the revised cut-off value for urinary anabasine compared to the current cut-off value.

Urinary anabasine is helpful for differentiating tobacco use in patients with nicotine replacement therapy [27]. However, the risk of misclassification by anabasine is less known. Among the positive samples with a 2.0 ng/mL cut-off value for urinary anabasine, about 30% were negative for nicotine or its metabolites, and 25% of the samples were below the LLMI in our study (Appendix A). This finding is in contrast to a previous study [28], where among the positive samples with a 3.0 ng/mL cut-off value for urinary anabasine, 0.4% were negative for nicotine metabolites. When using the same 3.0 ng/mL cut-off value, our data showed that 3.7% of samples were below the LLMI of urinary cotinine. Of these (568/15,195) samples, urinary anabasine distribution varied from 3.0 to 111.4 ng/mL.

The source of anabasine may be related to other foods or herbal medications rather than tobacco products [28]. For example, *Alangium platanifolium*, a small tree, is native to east Asia and is used for food and medicine [33]. The plant contains anabasine, and an experimental study on the extraction of DL-anabasine from *Alangium platanifolium* root has been reported [34]. Therefore, our data suggest that geological and cultural differences in anabasine intake should be considered for medical decisions to prevent misclassification of Korean patients with nicotine replacement therapy.

Our study has several limitations. Due to the limited authorization to use KNHANES data, including the age and tobacco-related questionnaire, we cannot distinguish between pediatrics and adults, while they may have different metabolic rates for nicotine metabolism. In addition, we cannot establish reference intervals for TNE3 and cannot validate the cut-off values of each biomarker for the same reason. To overcome these limitations, further study should be performed.

## 4. Materials and Methods

### 4.1. Sample Collection

A total of 19,874 urine samples were collected from anonymous Koreans over six years of age and delivered to Samsung Medical Center as frozen samples from the Korea Disease Control and Prevention Agency from January 2019 to October 2021 as a part of the 8th KNHANES (2019–2021). Samples were thawed and stored under refrigeration until preparation.

### 4.2. LC-MS/MS Analysis

The detailed assay protocol, including sample preparation, was conducted as previously published [18]. Briefly, 90 µL acetonitrile, 30 µL urine sample, and 5 µL deuterated ISs (co-tinine-d3, nicotine-d4, nornicotine-d4, 3-OH cotinine-d3, and anabasine-d4) were added to an Eppendorf tube. The mixture was centrifuged at 15,000 rpm for 10 min, and 30 μL of the supernatant was carefully transferred to another tube, which was then diluted with 120 µL of distilled water. A total of 7 µL of the sample solution was injected directly into an LC–MS/MS autosampler. For chromatographic separation, we used a Kinetex EVO C18 column (2.1 mm × 150 mm, 5 µm; Phenomenex, Torrance, CA, USA). The mobile phase was a gradient of (A) 30 mM ammonium bicarbonate and (B) acetonitrile at a flow rate of 0.4 mL/min. Gradient elution started with 91% A for 1.6 min, followed by a step to 85% A within 0.1 min, held for 3 min, and reequilibrated at 91% A from 4.7 min to 6.5 min. LC–MS/MS analysis was performed using a Waters XEVO TQ-S tandem quadrupole mass spectrometer (Waters Corporation, Milford, MA, USA) equipped with an Acquity UPLC system (Waters Corporation). The multiple reaction monitoring (MRM) transitions were as follows: cotinine (*m*/*z* 177.2 → 98.1 and 177.2 → 146.1), cotinine-d3 (*m*/*z* 180.2 → 101.1), nicotine (*m*/*z* 163.2 → 84.2 and 163.2 → 132.2), nicotine-d4 (*m*/*z* 167.3 → 134.2), nornicotine (*m*/*z* 149.1 → 130.2 and 149.1 → 80.1), nornicotine-d4 (*m*/*z* 153.2 → 134.2), 3-OH cotinine (*m*/*z* 193.2 → 134.2 and 193.2 → 80.1), 3-OH cotinine-d3 (*m*/*z* 196.2 → 134.2), anabasine (*m*/*z* 163.3 → 94.2, 163.3 → 92.0, and 163.3 → 146.3), anabasine-d4 (*m*/*z* 167.3 → 150.3). All acquisition methods used the following parameters: capillary voltage at 3.0 kV, source temperature at 150 °C, desolvation temperature at 450 °C, desolvation gas flow at 700 L/h, cone gas flow at 150 L/h, nebulizer pressure at 5.0 bar, and collision gas flow at 0.15 mL/min. The analysis time was approximately 6.5 min per sample. To determine precision and accuracy, four levels of spiking pooled urine with working solutions of each biomarker (2, 20, 100, and 1000 ng/mL for nicotine, nornicotine, and anabasine; 5, 50, 250, and 2500 ng/mL for cotinine; and 10, 100, 500, and 5000 ng/mL for 3-OH cotinine) were prepared. Our method was validated according to the standard procedure and guidelines [35,36,37,38,39]. The analytical performance of our measurement system is summarized in Appendix A. The LLMI was 5 ng/mL for 3-OH cotinine and 1 ng/mL for the other four biomarkers.

### 4.3. Data Analysis

The measured values of the five biomarkers were log-transformed due to skewed distributions. The value below the LLMI was replaced by the value of the LLMI divided by √2, and the value was considered “not detected” in this study. TNE3 was calculated by the molar sum of nicotine and two major metabolites, cotinine and 3-OH cotinine, without creatinine correction [13]. The correlation between the various biomarkers was examined by Spearman’s rank correlation. Referring to the study by Moyer et al. [19], 20 ng/mL for nicotine and cotinine, 50 ng/mL for 3-OH cotinine, and 2 ng/mL for nornicotine and anabasine were established as the cut-off values for the five biomarkers to distinguish between current smokers and non-current smokers. The degree of agreement between various biomarkers compared to urinary cotinine was quantified by Cohen’s kappa. Statistical Package for the Social Sciences (SPSS) version 25.0 (IBM Corporation, Armonk, NY, USA) and GraphPad Prism version 9.1.2. (GraphPad Software, La Jolla, CA, USA) were used for statistical analyses and graphing, respectively. Statistical significance was set at *p* < 0.05.

## 5. Conclusions

Taken together, our study shows that the single measurement of urinary cotinine has comparable classification results to differentiate between current smokers and non-current smokers compared to the analysis of multiple nicotine metabolites, or TNE3. However, measurements of multiple nicotine metabolites and TNE3 could be useful for monitoring exposure to low-level or secondhand smoke and determining individual differences in nicotine metabolism. Furthermore, geological or cultural factors of anabasine intake should be considered for the use of urinary anabasine for differentiating tobacco use from patients with nicotine replacement therapy.

## Figures and Tables

**Figure 1 molecules-28-07685-f001:**
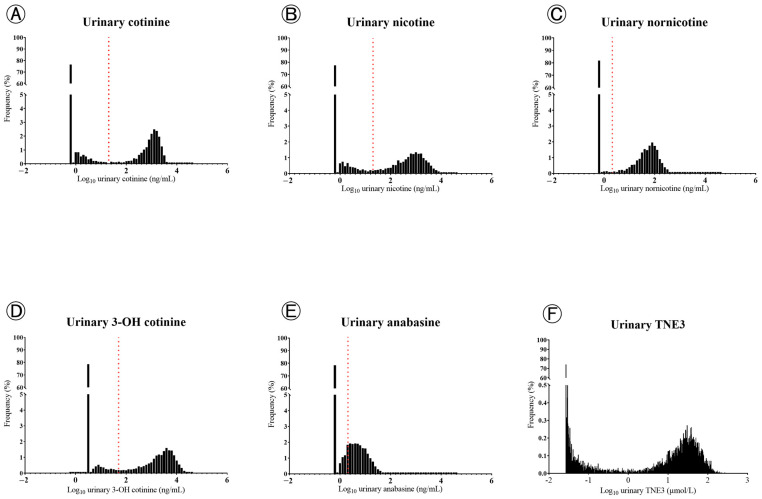
Distribution and cut-off values of various biomarkers related to smoking exposure: (**A**) cotinine; (**B**) nicotinine; (**C**) nornicotine; (**D**) 3-OH cotinine; (**E**) anabasine; and (**F**) TNE3.

**Table 1 molecules-28-07685-t001:** The results of measurement for five urinary biomarkers related to smoking exposure using 19,874 urine samples.

Biomarkers	Not Detected	Detected	Positive	Concentration	
	*n* (%)	*n* (%)	*n* (%)	(Mean ± SD; ng/mL)	Range (ng/mL)
Cotinine	15,195 (76.5%)	4679 (23.5%)	3682 (18.6%)	221.2 ± 578.1	<1.0–4907.4
Nicotine	15,400 (77.5%)	4474 (22.5%)	3460 (17.4%)	214.5 ± 776.5	<1.0–13,806.4
Nornicotine	16,232 (81.7%)	3642 (18.3%)	3434 (17.3%)	14.3 ± 41.3	<1.0–766.5
3-OH cotinine	15,649 (78.7%)	4225 (21.3%)	3308 (16.6%)	715.3 ± 2307.5	<5.0–39,604.5
Anabasine	15,588 (78.4%)	4286 (21.6%)	2624 (13.2%)	2.2 ± 5.1	<1.0–161.1
TNE3 (µmol/L)				6.3 ± 18.0	<0.1–268.5

3-OH cotinine, trans-3′-hydroxycotinine; TNE3, total nicotine equivalents three (the molar sum of nicotine, cotinine, and 3-OH cotinine). Furthermore, 20 ng/mL for nicotine and cotinine, 50 ng/mL for 3-OH cotinine, and 2 ng/mL for nornicotine and anabasine are the cut-off values of five biomarkers for distinguishing between current smokers and non-current smokers.

**Table 2 molecules-28-07685-t002:** Comparison of the detection status of five biomarkers.

Urinary Biomarkers	Cases	Cotinine		Nicotiine		Nornicotine		3-OH Cotinine		Anabasine	
		Detected	Not Detected	Detected	Not Detected	Detected	Not Detected	Detected	Not Detected	Detected	Not Detected
		(4679)	(15,195)	(4474)	(15,400)	(3642)	(16,232)	(4225)	(15,649)	(4286)	(15,588)
Cotinine	Detected			4116	563	3625	1054	4122	557	3081	1598
	Not detected			358	14,837	17	15,178	103	15,092	1205	13,990
	*κ*			0.869	(0.861–0.878)	0.838	(0.829–0.847)	0.905	(0.897–0.912)	0.597	(0.583–0.610)
Nicotine	Detected	4116	358			3621	853	3892	582	3078	1396
	Not detected	563	14,837			21	15,379	333	15,067	1208	14,192
	*κ*	0.869	(0.861–0.878)			0.973	(0.968–0.977)	0.865	(0.857–0.874)	0.619	(0.605–0.632)
Nornicotine	Detected	3625	17	3621	21			3594	48	2957	685
	Not detected	1054	15,178	853	15,379			631	15,601	1329	14,903
	*κ*	0.838	(0.829–0.847)	0.865	(0.856–0.874)			0.893	(0.885–0.900)	0.683	(0.670–0.696)
3-OH cotinine	Detected	4122	103	3892	333	3594	631			3013	1212
	Not detected	557	15,092	582	15,067	631	15,018			1273	14,376
	*κ*	0.905	(0.897–0.912)	0.865	(0.857–0.874)	0.893	(0.885–0.900)			0.628	(0.615–0.642)
Anabasine	Detected	3081	1205	3078	1208	2957	1329	3013	1273		
	Not detected	1598	13,990	1396	14,192	685	14,903	1212	14,376		
	*κ*	0.597	(0.583–0.610)	0.619	(0.605–0.632)	0.683	(0.670–0.696)	0.628	(0.615–0.642)		

3-OH cotinine, trans-3′-hydroxycotinine; *κ*, Cohen’s kappa. Furthermore, 5 ng/mL for 3-OH cotinine and 1 ng/mL for cotinine, nicotine, nornicotine, and anabasine are the lower limits of the measuring interval.

**Table 3 molecules-28-07685-t003:** Comparison of the classification of smoking status based on five biomarkers.

Urinary Biomarkers	Cases	Cotinine		Nicotine		Nornicotine		3-OH Cotinine		Anabasine	
		Positive	Negative	Positive	Negative	Positive	Negative	Positive	Negative	Positive	Negative
		(3682)	(16,192)	(3460)	(16,414)	(3567)	(16,440)	(3574)	(16,566)	(3566)	(17,250)
Cotinine	Positive			3437	245	3550	113	3519	163	2641	1041
	Negative			23	15,172	17	15,178	55	15,140	925	14,270
	*κ*			0.954	(0.948–0.959)	0.978	(0.974–0.982)	0.963	(0.958–0.968)	0.664	(0.651–0.678)
Nicotine	Positive	3437	23			3434	26	3308	152	2624	836
	Negative	245	16,169			133	16,281	266	16,148	942	15,472
	*κ*	0.954	(0.948–0.959)			0.973	(0.968–0.977)	0.968	(0.964–0.973)	0.693	(0.679–0.706)
Nornicotine	Positive	3550	17	3434	133			3417	150	2636	931
	Negative	132	16,308	26	16,414			157	16,283	930	15,510
	*κ*	0.978	(0.974–0.982)	0.973	(0.968–0.977)			0.948	(0.942–0.953)	0.683	(0.669–0.696)
3-OH cotinine	Positive	3519	55	3308	266	3417	157			2562	1012
	Negative	163	16,403	152	16,414	150	16,416			1004	15,562
	*κ*	0.963	(0.958–0.968)	0.968	(0.964–0.973)	0.948	(0.942–0.953)			0.657	(0.643–0.671)
Anabasine	Positive	2641	925	2624	942	2636	930	2562	1004		
	Negative	1041	16,209	836	16,414	931	16,319	1012	16,238		
	*κ*	0.664	(0.651–0.678)	0.693	(0.679–0.706)	0.683	(0.669–0.696)	0.657	(0.643–0.671)		

3-OH cotinine, trans-3′-hydroxycotinine; *κ*, Cohen’s kappa. Furthermore, 20 ng/mL for nicotine and cotinine, 50 ng/mL for 3-OH cotinine, and 2 ng/mL for nornicotine and anabasine are the cut-off values of five biomarkers for distinguishing between current smokers and non-current smokers.

**Table 4 molecules-28-07685-t004:** Combinations of the five biomarkers for detection status.

Detection Status		
Cases	Number	%
Not detected	13,598	68.4
Single biomarker detectable	1788	9.0
Cotinine	275	1.4
Nicotine	281	1.4
Nornicotine	14	0.1
3-OH cotinine	76	0.4
Anabasine	1142	5.7
Two biomarkers detectable	529	2.5
Cotinine + Nicotine	187	0.9
Cotinine + Nornicotine	1	0.0
Cotinine + 3-OH cotinine	224	1.1
Cotinine + Anabasine	35	0.2
Nicotine + 3-OH cotinine	20	0.1
Nicotine + Anabasine	55	0.3
Nornicotine + Anabasine	2	0.0
3-OH cotinine + Anabasine	5	0.0
Three biomarkers detectable	320	1.6
Cotinine + Nicotine + Nornicotine	20	0.1
Cotinine + Nicotine + 3-OH cotinine	242	1.2
Cotinine + Nicotine + Anabasine	28	0.1
Cotinine + Nornicotine + 3-OH cotinine	4	0.0
Cotinine + 3-OH cotinine + Anabasine	1	0.0
Nicotine + Nornicotine + 3-OH cotinine	1	0.0
Nicotine + 3-OH cotinine + Anabasine	24	0.1
Four biomarkers detectable	695	3.5
Cotinine + Nicotine + Nornicotine + 3-OH cotinine	645	3.2
Cotinine + Nicotine + Nornicotine + Anabasine	11	0.1
Cotinine + Nicotine + 3-OH cotinine + Anabasine	39	0.2
Five biomarkers detectable	2944	14.8
Total	19,874	100.0

3-OH cotinine, trans-3′-hydroxycotinine. Furthermore, 5 ng/mL for 3-OH cotinine and 1 ng/mL for cotinine, nicotine, nornicotine, and anabasine are the lower limits of the measuring interval.

## Data Availability

The data are not publicly available due to limited authorization to use KNHANES data.

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
