# Peer review of "Simultaneous Measurement and Distribution Analysis of Urinary Nicotine, Cotinine, Trans-3′-Hydroxycotinine, Nornicotine, Anabasine, and Total Nicotine Equivalents in a Large Korean Population"

_molecules, 2023, doi:10.3390/molecules28237685_

Round 1

Reviewer 1 Report

Comments and Suggestions for Authors

This paper focuses on tobacco exposure verification and the authors applied the analytical method established by them to 19,874 urine samples. The authors concluded that single measurement of urinary cotinine is simple and has comparable performance to differentiate between current smokers and non-smokers. However, the authors also mentioned that measurement of multiple nicotine metabolites and TNE3 could be useful for monitoring exposure to low-level or secondhand smoke exposure. This paper does not propose new analytical method, but the results of large-scale analysis might be useful for those dealing with biological samples to evaluate tobacco exposure level. For that reason, I think this manuscript might be published, but major revision would be needed before full consideration.

General comments and questions:

1. I could not fully understand the difference between Table 2 and Table 3. Are both of those tables needed or only one table is enough to describe the result? Please clarify that point.

2. As a paper which describes the analytical results, the detailed analytical conditions (instrumental conditions of LC-MS/MS such as column, eluent, and ionization parameters) would be needed even though they were almost the same with the previous paper.

3. In Table 1, the SDs are large, and that might show that the concentration ranges were broad. I recommend the authors to add the concentration ranges to Table 1.

4. In Figure 3, some of the analytes shows bimodal distribution and others does not show those behavior. What do you think is the reason for this difference?

5. For supporting information, the contents should be ordered according to the order they are mentioned in the manuscript for readers’ understanding.

Reviewer 2 Report

Comments and Suggestions for Authors

The article “Simultaneous Measurement and Distribution Analysis of Urinary Nicotine, Cotinine, trans-3'-hydroxycotinine, Nornicotine, Anabasine, and Total Nicotine Equivalents in a Large Korean Population” aims to assess the agreement between single urinary cotinine and multiple nicotine metabolites in a large number of samples from a national survey. The article appears methodologically correct. However, its novelty is limited, as previous studies have explored correlations between nicotine metabolites and nicotine. For instance, a prior article by the same group titled “A Simple and High-Throughput LC-MS-MS Method for Simultaneous Measurement of Nicotine, Cotinine, 3-OH Cotinine, Nornicotine and Anabasine in Urine and Its Application in the General Korean Population”: https://pubmed.ncbi.nlm.nih.gov/33231618/ covered similar ground.

The article speculates on several points due to a lack of patient data, which the authors acknowledge as a limitation. Nevertheless, the conclusion suggests that cotinine, the most frequently measured metabolite for detecting smokers, is the best and sufficient marker. The study's significant sample size and the analysis of metabolite concentration distribution within this group are notable strengths.

In the introduction, there's a lack of information regarding the state of the art and the knowledge gap. In my view, the conclusions „Our study shows that the single measurement of urinary cotinine is simple and has comparable performance to differentiate between current smokers and non-smokers relative to the measurement of multiple nicotine metabolites. However, measurement of multiple nicotine metabolites and TNE3 could be useful for monitoring exposure to low-level or secondhand-smoke exposure and for determining individual differences in nicotine metabolism.” are inadequately supported. How was this validated without patient smoking status data? Which marker was used as a reference? TNE? Please provide justification.

Detailed comments:

1. What is "TNE 3" (line 24)? Please clarify.

2. Page 3, lines 82-87: What is the text about? There seems to be an editorial issue, and LLMI and IQR are unnecessary.

3.Lines 97-99: This sentence sounds odd.

4. Table 2: Include cut-off values.

5. Line 160: Is 4 the reference?

6. Line 172: Be more specific. Which one?

7.What criteria were met to use the Pearson test to assess correlations?

8.Table S1: "ME" should be presented as %RSD on different lots of urine.

9. Table S1: For what concentration level was validation conducted (precision and accuracy)? Following EMA guidelines, it should be performed at three concentration levels plus LLOQ

Round 2

Reviewer 1 Report

Comments and Suggestions for Authors

The authors corrected the manuscript according to the reviewers’ comments and I think the manuscript was improved compared with the former version. However, I think there remains some points to be corrected. After those improvements, I think this manuscript could be considered for publication.

1. Table 2:  “Nicotinine” should be replaced by “Nicotine”.

2. Table 3:  “Nicotinine” should be replaced by “Nicotine”.

3. L117: “detection status status” → ”detection status”

4. Supplementary Table 1:  “Nicotinine” should be replaced by “Nicotine”.

5. Supplementary Table 2:  “Nicotinine” should be replaced by “Nicotine”.

6. Supplementary Table 3:  “Nicotinine” should be replaced by “Nicotine”.

Reviewer 2 Report

Comments and Suggestions for Authors

I am satisfied with the corrections. 
